# Where did the river go? Testing the hypothesis of rivers discharging into the Gulf of Sirt (East Mediterranean) during the late Pleistocene

Barbara Mauz[1,2*], Esam Abdulsamad[3], Saleh Emhanna[4], Noureddine Elmejdoub[5], Moez Mansoura[6], Michael Rogerson[7]

1 School of Environmental Sciences, University of Liverpool, Liverpool, United Kingdom, 2 Department of Environment and Biodiversity, University of Salzburg, Salzburg, Austria, 3 Earth Sciences Department, University of Benghazi, Benghazi, Libya, 4 Colllege of Engineering, University of Ajdabiya, Ajdabiya, Libya, 5 Laboratory: Applied Hydrosciences, Higher Institute of Water Sciences and Techniques, University of Gabès, Gabès, Tunisia, 6 Office National des Mines, Tunis, Tunisia, 7 Department of Geography and Environmental Science, Northumbria University, Newcastle upon Tyne, United Kingdom

* barbara.mauz@plus.ac.at

## Abstract

Large quantities of freshwater supplied by rivers are, amongst other factors, required to slow down deep-water ventilation and allow sapropels to form. Spatial distribution of sapropels in the East Mediterranean as well as its thermohaline circulation point to rivers reaching the African coast to the west of the Nile. Here we study the coastal plain of the Gulf of Sirt (Libya) to find evidence for rivers. Using field survey, laboratory analyses on coastal samples and published geological data from wells and surface mapping we find a carbonate-rich, clastic-starved Gulf coast prevailing during MIS 5 and early MIS 4. The coastal plain is a flat and featureless Pliocene surface lacking evidence for a large-scale allogenic river but showing some water discharge in a desert depression situated ca 200 km inland. While we have to conclude that no river reached the Gulf of Sirt during MIS 5, we found evidence for ponding of brackish water in the Chott El Jerid (Tunisia) and support the idea of a Irharhar – Chott water pathway.

## Introduction

In the East Mediterranean Sea sapropels (i.e., organic-rich beds intercalated in hemipelagic sediments) are not conceivable without substantial amount of freshwater discharged by large rivers [1]. The Nile is certainly an important river in this respect [1,2], but data [3,4] suggest that other north African rivers have supplied water to the Ionian Sea. The much increased north African hydrological budget is likely to have been caused by the northward shift of the African Monsoon domain as a response to the northward move of the Intertropical Convergence Zone during the boreal summer insolation anomaly [5]. This move is regarded as the driver of African humid

**Data availability statement:** All relevant data are within the paper and its Supporting Information files.

**Funding:** The National Authority for Scientific Research (Libya) funded field work in Libya . Field work in Tunisia (Chott area) was supported by Leverhulme Grant IN-2012-113 (awarded to M Rogerson).

**Competing interests:** The authors have declared that no competing interests exist.

periods that have generated green corridors in the Sahara desert suitable for human migration [6–10]. The ages of most sapropels coincide with the precession-minimum induced intensification and expansion of the African Monsoon [11], so it is reasonable to assume that a southern source of freshwater supported the formation of most sapropels. Rivers crossing the Saharan desert and discharging into the Gulf of Sirt (Fig 1), hence to the west of the Nile, namely the Sahabi and the Kufrah rivers (Fig 2), have been invoked in a remarkable number of publications on the subject [5,12–16]. If the African Monsoon domain expanded to the north during one or more precession minimum periods and triggered freshwater input into the Ionian Sea, perennial or intermittent rivers carrying Saharan clastic sediment would have crossed the Sirt coastal plain. We tested this hypothesis by investigating the Neogene geology of the Gulf of Sirt coastal plain. Our ground-truthing approach aimed at finding evidence for fluvial activity and freshwater ponding in the coastal plain.

### Constraining freshwater supply

Freshwater flux to the eastern Mediterranean is inferred from the deep-sea sediment cores (see Fig 3 for location of those boreholes that are situated close to the Gulf of Sirt). These cores cover different time intervals, mostly encompassing one or more sapropels. The analysis of the hemipelagic sediments and sapropels include minerals, stable and radiogenic isotopes, bulk and trace element composition and grain size. Plots of $^{87}Sr/^{86}Sr$ versus $\varepsilon Nd$ allow separating Tunisian and Saharan dust (both transported by the Ghibli winds) from Libyan soil (transported by rivers) because $^{87}Sr/^{86}Sr$ is lower in the soils [19] compared to the dust [20]. Elemental data and the Ba/Al ratio [3], diluted eNd values [13,15] and grain-size end-member data [15], all indicate increased fluvial input best explained with a three endmember mixing model encompassing Nile/Aegean Sea, Saharan dust and Libyan soil [3]. Finally, there is a clear correlation between $\delta^{18}O$ anomalies and sapropels and these anomalies are strongest in the western Ionian Sea [21]. To generate these geochemical anomalies the integrated freshwater flux must include runoff from the Mediterranean borderlands that exceeds evaporation. LOVECLIM model predicts ~400 mm/a rainfall over the Libyan coast during MIS 5e and ~300 mm/a during the entire MIS 5 [15]. Satellite imagery [12,22] and hydraulic modelling [14] support the evidence from the deep sea by reconstructing or simulating river courses that cross the Sahara desert (Fig 2).

Recording north Africa fluvial provenance in deep-sea cores largely depends on surface water circulation and wind regime over the Gulf of Sirt. Today, the Sidra gyre is located north of the Misrata-Tripoli promontory (Fig 3; [23]). In autumn/winter the gyre is contracted and rests off the 200 m isobath [24] and the main Atlantic-Tunisian current runs along the shelf edge in SE direction pumping sediment from the sand-rich Tunisian shelf towards the Gulf. During the remaining year the anticyclonic gyre expands onto the shelf and the shelf current runs in the opposite direction (see pink arrows in Fig 3). Thermodynamic water circulation must have been subdued or suspended when large water masses were supplied to the Ionian Sea as inferred for the sapropel S1 interval [25].

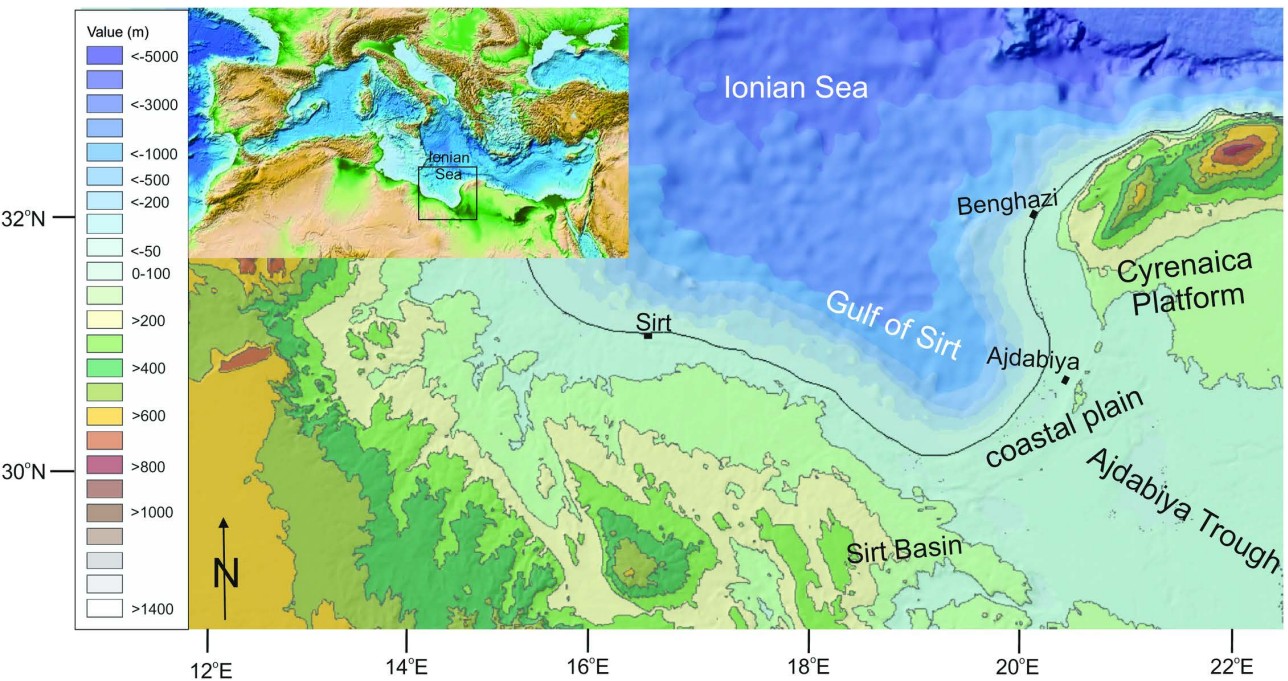

**Fig 1. The Gulf of Sirt and surrounding area; the Cretaceous-Miocene Cyrenaica platform to the north east and east, the Ajdabiya Trough and the coastal plain to the south, the Cretaceous-Miocene hill landscape of the Sirt Basin to the west.** See inset for location within the Mediterranean Sea. Background data reprinted from NOAA National Centers for Environmental Information 2022: ETOPO 2022 15 Arc-Second Global Relief Model, https://doi.org/10.25921/fd45-gt74 under a CC BY licence.

If freshwater was supplied to the Ionian Sea during African Monsoon belt expansion times, the Nubian Sandstone Aquifer System (NSAS) must have been affected by enhanced rainfall in the Erdi, Ennedi and Tibesti mountains [28–30]. The stable isotopes of the fossil NSAS waters are depleted and very light ($\delta^{18}$O: −10‰ to −11‰, $\delta^{18}$D: −72‰ to −80‰; [30]) with notable constant values across the entire aquifer area and across the deeper and shallower aquifers [31]. In contrast, the modern groundwater in the Sahelian zone is less depleted and heavier than the NSAS waters ($\delta^{18}$O: −2.2‰ to −5.8‰, $\delta^{18}$D:-10‰ to − 37.1‰; [32]). A broad consensus is that for the Saharan zone the isotope signature indicates air masses originating from the Atlantic which lose their heavy fraction through condensates when travelling from west to east [30,33,34] while the Sahelian zone shows the signature of the Indian monsoon in the east and the Atlantic monsoon in the west [30,32]. The differences between Sahara and Sahel zones seem significant, notwithstanding altitude, temperature, age and other parameters controlling the isotope ratios. Also, timing of peak discharge from the Nubian aquifer, recorded in travertines in Egypt, does not align with the timing of sapropels [35]. Thus, the Nubian aquifer appears to be not dominated by an expanded African monsoon belt but was controlled by a number of regional and spatially variable forcings.

## Finding evidence

Our study area is the low-gradient downstream part of an allogenic river originating from the Erdi, Ennedi and Tibesti mountains. We assume precession minimum-induced seasonality of enhanced rainfall events in these mountains which fed the trans-Saharan rivers. On the way to the coast, the river could have gained additional water from a sub-alluvial aquifer and from hillslope runoff in response to mean annual coastal rainfall estimated to 280–400 mm/year by the LOVECLIM climate model [15]. Hydrologically, such a river is characterised by seasonal flood with a multi-peak hydrograph depending on the magnitude of Mediterranean storm tracks. We are guided

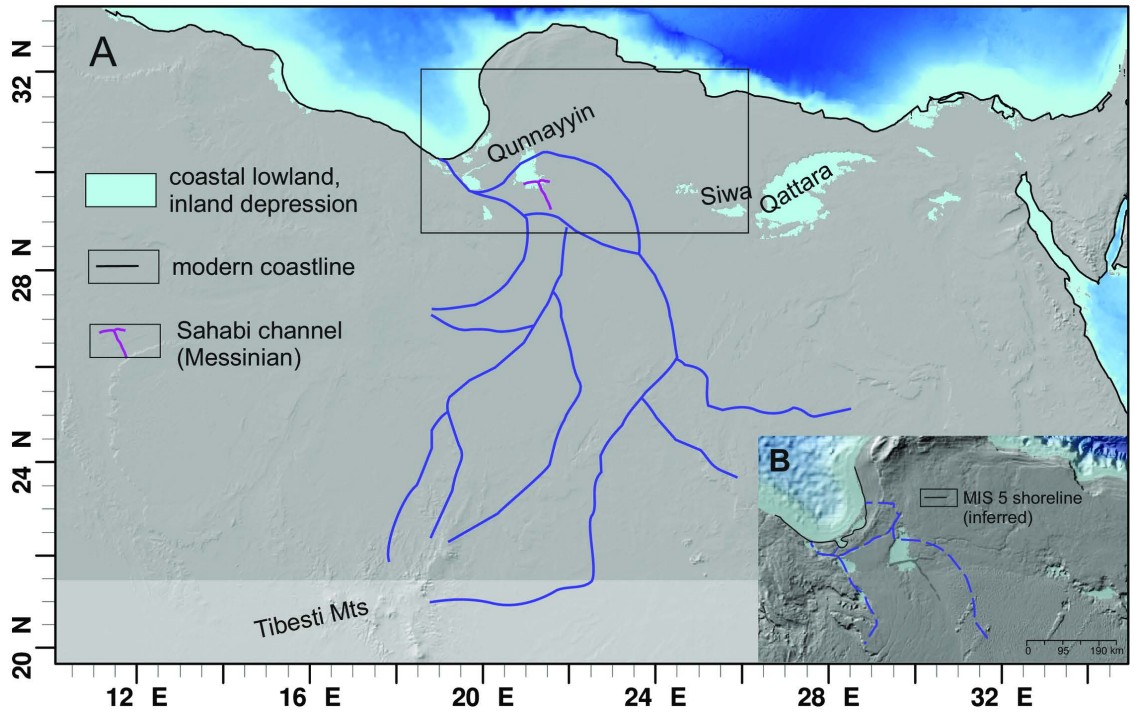

**Fig 2. The MIS 5 coast during sea-level highstand and inland depressions in northeast Africa.** A – the hypothetical river courses adopted from [5]. B – Details of the river course in the study area following [14,17]. The Messinian Sahabi channel [18] is shown for comparison. Background data reprinted from NOAA National Centers for Environmental Information 2022: ETOPO 2022 15 Arc-Second Global Relief Model. https://doi.org/10.25921/fd45-gt74 under a CC BY licence.

by the combined hydrological-hydraulic model simulation [14] which raises the possibility of looking for remains of intermittent to near-perennial streams and short-lived inundation of the coastal plain. Given silt and sand dominated sediment load, the river channel should be characterised by a high width to depth ratio identified through relict, relatively straight lines marking the near-planar surface of a wide stream filled with fining upward sand, silt and mud against the older background surface sediment. The low-gradient coastal plain could show thinly bedded deposits composed of poorly sorted sand and silt, eventually separated by reactivation surfaces and indurated by secondary evaporites such as gypsum and carbonate. After a flood episode, punctuated wind deflation of the plain, moisture concentration in sabkhas and enhanced groundwater sapping in depressions in correspondence with high sea level should have occurred. These processes would be recorded by lunette dunes bordering the downwind side of the formally inundated area and by evaporites in sabkhas. In enclosed depressions waterfall erosion and corresponding tufas, as well as mass wasting deposits may be found. On the coast, sediment would be well-winnowed and dominated by fluvially supplied material which would have been reworked on the microtidal, wind-dominated nearshore. Backshore sabkhas would have formed behind the foredune and the foredune itself would be dissected at a river outflow.

Finding these remains preserved and unburied depends on dryland processes such as wind deflation and sand mobilisation and on the presence of obstacles capable of holding material on the ground. On the flat coastal plain most loose material was probably reworked to dunes, but indurated deposits and the geomorphic imprint of surface water is typically well preserved [27]. The lack of findings would give rise to speculate about transmission losses promoting spatial and temporal intermittency of river flow.

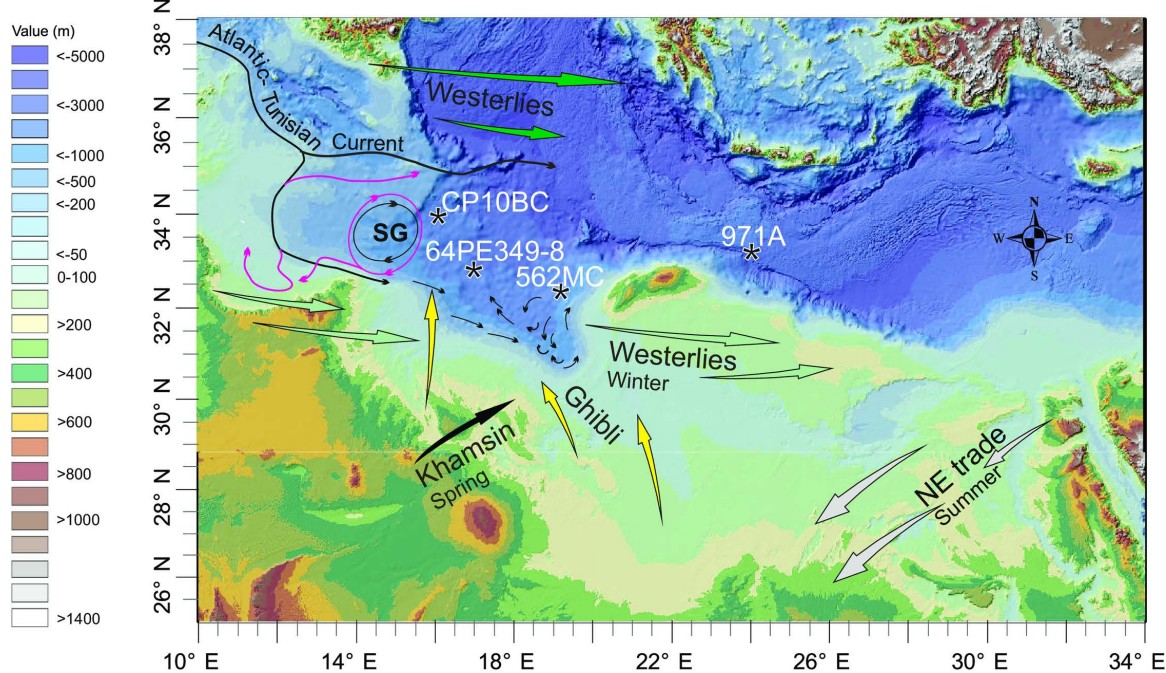

**Fig 3. Modern East Mediterranean surface-water circulation along the Tunisian-Libyan coast after [23,26], regional wind regime and the location of the boreholes (*CP10BC*: 1501 m water depth [3]; *971A*: 2026 m water depth [13]; *64PE 349-8*: 2095 m water depth [15]; 562MC: 1390 water depth [27]).** SG = Sidra Gyre. The Ghibli winds (yellow arrows) blow northwards on the east flank of Mediterranean low-pressure cells. The Khamsin winds (black arrow) blow over land and push dry sediment eastward. Also shown is the summer position of NE trade winds (white arrows) and summer/winter position of westerlies (green and light green arrows). Background data reprinted from NOAA National Centers for Environmental Information 2022: ETOPO 2022 15 Arc-Second Global Relief Model. https://doi.org/10.25921/fd45-gt74 under a CC BY licence.

## Methods

Our approach comprised field work, laboratory-based analytical work on samples collected in the field and literature-based work to compile data relevant to our ground-truthing approach.

Field work included standard procedures of surveying, logging and sampling outcrops to find evidence for water ponding, intermittent streams and post-event remains such as lunette dunes (see Supporting Information about field work permission). Laboratory work focused on petrographic thin-section analysis of the coastal deposits using standard petrographic microscope procedure. Sediment classification followed [36]. Where quartz grains were available and age was expected to be < 150 ka, optical dating was employed (for details of optical dating see Supporting Information).

The flat and not dissected coastal plain required the study of well data [37], the geological map [38] and hydrogeological literature to better constrain stratigraphy, structural configuration of the Sirt basin and groundwater pathways.

## Results

The Ajdabiya Trough (Fig 1) is filled with around 4 km post-Cretaceous sediment [39], the uppermost ca 600 m of which are assigned to the Mio-Pliocene (Fig S1 in S1 File). The northeast of the Trough is filled by up to 60 m Pliocene coastal-sabkha sediments (Qarat Weddah Fm), which dominates the surface area of the Trough (Fig 4) and represents the Sirt coastal plain. The elevated Moho depth beneath the Sirt basin [40] and the thick Neogene sediment sequence indicate crustal thinning and continuous subsidence estimated to 60−110 m in 1 Ma (ca 0.06–0.11 mm/a) during the past 23 Ma [37,40].

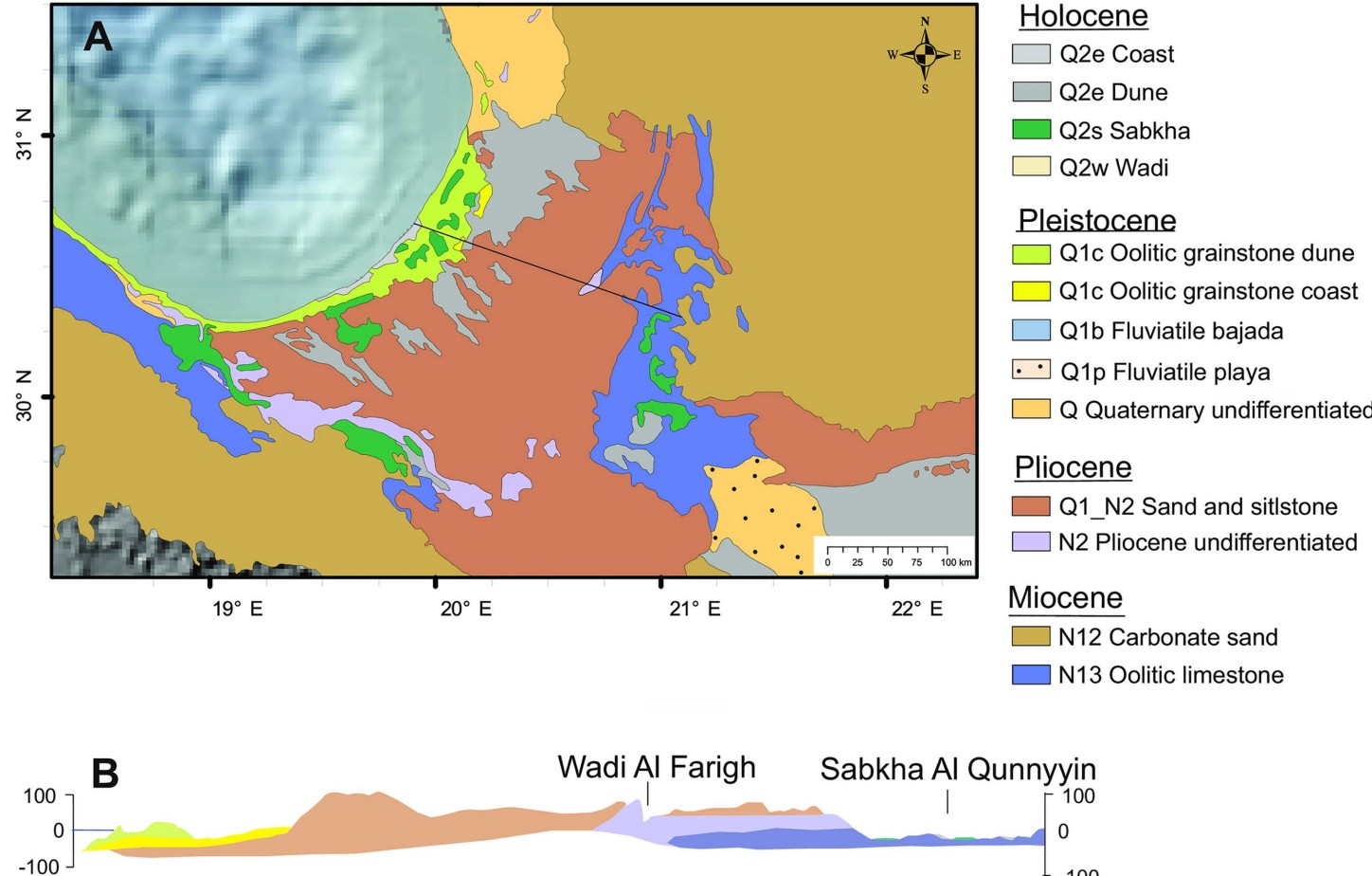

**Fig 4. Geological setting of the Ajdabiya Trough; A – Geological map modified from [38] showing surface strata; B – Cross section (see black line in A for location).** Background data reprinted from NOAA National Centers for Environmental Information 2022: ETOPO 2022 15 Arc-Second Global Relief Model. https://doi.org/10.25921/fd45-gt74 under a CC BY licence.

The Pliocene is overlain by 1–3 m Plio-Pleistocene calcrete crust along the Cyrenaica escarpment, by coastal oolitic grainstone (Ajdabiya Fm; Fig S4 in S1 File) and aeolian oolitic grainstone near the modern coast. The ooids exhibit aragonitic tangential cortex laminae. Ooids transported by wind are small (~200 µm) with an abrasion-induced diminished cortex compared to their original counterpart (~500 µm; Table S1 in S1 File). The deposits do not contain any siliciclastic components until they reach the margins of the Sirt bay where they pinch out and mix with bio-siliciclastic sand. The oolitic dune forms a coastal dune belt interfingering with 2–3 m silty clay sabkha deposits which fill interdune lows (Fig S5 in S1 File). The top of the dune belt is 62±2 ka old (for details see Supporting Information). To the east, the Cyrenaica escarpment is covered by several meters of quartz sand forming Holocene linear dunes. All post-Miocene deposits, apart from the coastal dune belt and the Holocene linear dunes, are horizontally bedded, enhancing the expression of a flat- and low-lying land surface with no relief (Fig S3 photo in S1 File).

## Surface and subsurface features in the Ajdabiya Trough

The surface of the Ajdabiya Trough is a gently rising, flat and featureless plain extending from the coastline ca 200 km inland up to an altitude of around 130 m. The planar surface, partly covered by calcrete crust, reflects the Miocene

carbonate platform modified by the post-Messinian transgression by which the Messinian incisions were filled and levelled [41]. One of these Messinian incisions is the Sahabi channel (Fig 2). Exceptions to this monotonous landscape are several depressions, the wadi Al Farigh and the Ajdabiya-Tobrouq ridge (Fig 5).

Six depressions, today isolated features, occur in the Ajdabiya Trough on altitudes between 0 m and 130 m and with bottom elevations near or below mean sea level [38,42], (Fig 4B). The depressions are sagged into Pliocene strata and thus, started forming during or soon after the early Pliocene (see logs in Fig S2 in S1 File). Each of the western depressions is connected to a wadi or to the sea. The Qunnayyin depression, first connected to several eastern wadis draining the Cyrenaica platform, is connected to a fluvial plain in the south since post-Pliocene times (Fig 5). In the north an incision connects the depression with the sabkha Hamra and with the wadi Farigh. Besides two ca 3 m thick aeolian sand areas in the southwestern part representing 5–6 km$^3$ sand and silt, the depression's floor exhibit bedrock partly covered by thin sabkha deposits. No water runoff features such as cliff incisions are identified along its western steep margin.

The wadi Al Farigh runs subparallel to the coastline with bottom elevation lying below 50 m and generally decreasing to the west. The wadi is narrow and deep in its eastern part and widens downstream before it reaches the sabkha Kabirah (Fig 5). Some up to 5 m thick wadi deposits are found along its course [38]. Form, size and orientation of wadi incision indicate initiation during Messinian times and flow to the west since the Pliocene.

The 50–100 m high Ajdabiya-Tobrouq Ridge, composed of Pliocene silty limestone (Fig S6 in S1 File), also runs subparallel to the coastline. No fluvial incision was found in the eastern part of the ridge and no topographic break was identified in the digital elevation model (DEM; Fig 5). Wadi course and ridge structure reflect the westward shift of a post-Miocene structural deformation that gently arched the Pliocene deposits [37], evident by the thickening of the corresponding deposits towards the west, the break-up of the calcrete cover and the internal divide in the wadi channel.

The Messinian Sahabi river channel is today situated at around 400 m below sea level [37], where it is incised into late Miocene deposits and filled with Pliocene sand and silt bearing subtropical freshwater fauna (e.g., rhinoceros, giraffe, etc; [37]).

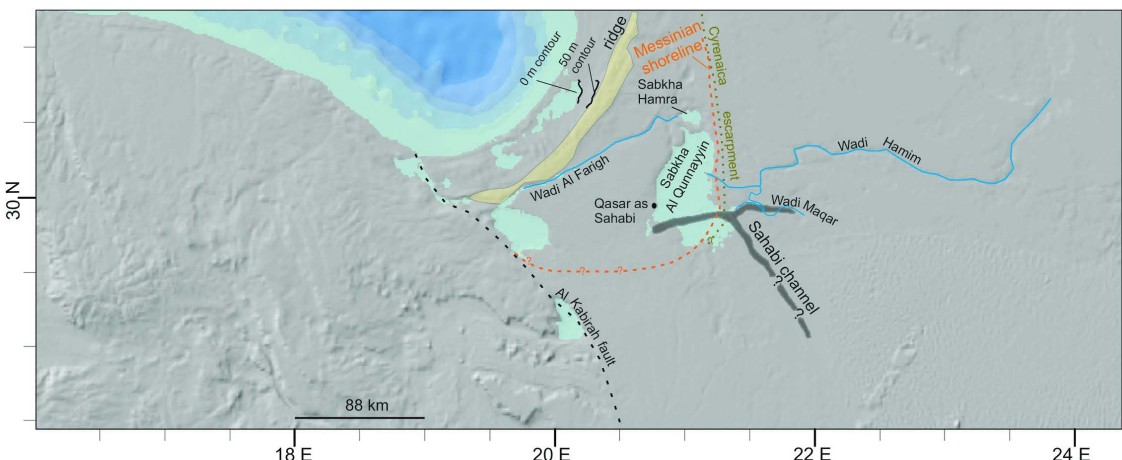

**Fig 5. DEM of the Ajdabiya Trough showing sabkha depressions (light blue), wadi courses (blue) and the Ajdabiya-Tobrouq Ridge (yellow).** Reconstructions follow [18] for the position of the Messinian Sahabi channel, [41] for the Cyrenaica escarpment and the Al Kabirah fault and [38] for the Messinian shoreline. Qasar as Sahabi is the key section for the mid-late Miocene succession [41]. See also Fig S6 in S1 File. Background data reprinted from NOAA National Centers for Environmental Information 2022: ETOPO 2022 15 Arc-Second Global Relief Model. https://doi.org/10.25921/fd45-gt74 under a CC BY licence.

## Discussion

### Arid shoreline and fluvial base level

It is clear from the stratigraphic-structural analyses that the Trough experienced spatio-temporal variation of the stress field, continuous subsidence of <1 cm/a (since Luetian (250–300 m) [39]) and younging of its surface towards the west. The Trough is filled by >30 m of Pliocene detrital carbonate sediment, clay and silt of shallow marine, coastal sabkha environment which fill the coastal plain rapidly and constrain shoreline retreat. Ongoing subsidence and simultaneous shoreline retreat make the Sirt coastal plain the base level for all post-Pliocene trans-Saharan rivers as inferred by previous studies {e.g., [1,6,12–15]). Deposits behind the modern coastline are associated with the MIS 5e sea-level highstand and Holocene aeolian sand transport. No deposits of post-Miocene floodplain and/or fluvial environments are present in the coastal plain including the Qunnayyin depression.

The coastal zone is dominated by oolitic grainstone deposits representing a foreshore environment with onlap geometry (Ajdabiya Fm) and backshore foredune environments in offlap geometry (Gargaresh Fm) on the innermost shelf (Figs S and S5 in S1 File). The ooids of the Sirt coast exhibit average characteristics (e.g., nucleus to cortex ratio) often described in literature (see Supporting Information for details). Ooids carry a record of their environment of formation (e.g., [43]) as sea-water chemistry and sea-surface temperature can be approximated through their size, cortex and mineralisation [44–46]. On the Sirt coast the oolitic deposits represent warm coastal water characterised by elevated alkalinity (~2400 umol kg$^{-1}$; [45]) and elevated salinity (typical for the east Mediterranean today) to allow cortex formation composed of aragonitic laminae. The deposits represent the MIS 5 sea-level highstand when sea level oscillated on a $10^3$ to $10^4$ time scale between around 7 m [47] and −50 m [48]. The maximum transgression, occurring 127–115 ka [49], is represented by the Ajdabiya Fm surfacing about 10 km behind the modern shoreline (Fig 4 signature Q1c). The maximum regression observed on land is represented by the top of the oolitic coastal dune yielding an OSL age of ~62 ka (MIS 4). Taking into account spatial and temporal data as well as composition and thickness of the oolitic deposits (Fig S2 in S1 File), an arid nearshore covered by alkaline seawater is inferred for most part of MIS 5 until MIS 4. During MIS 5e arid coastal conditions prevailed also to the east of the Sirt Gulf, on the Alexandria coast (Egypt) and on the Levant coast, and, during MIS 5a, also to the west, on the Libyan-Tunisian coast [50–52].

The Qunnayyin and Kabirah depressions (Fig 5) started forming in the early Pliocene. The semi-enclosed depressions must have been the base level for hillslope runoff water and for all post-Messinian allogenic rivers coming from south. Evidence for this is provided at the Qunnayyin depression: reworked and partly eroded caliche crust at its eastern margin (see log L1 in Fig S2 in S1 File) and mappable Quaternary fluvial sediment plume at its southern margin (see signature Q1p in Fig 4). This sediment plume is not linked to a stream but terminates at the Qunnayyin depression (Fig 6).

### The MIS 5 water course

During flood season the depressions could have formed confluent streams and coalesce with the wadi Al Farigh to ultimately generate an ephemeral floodplain on the Pliocene surface between the Ajdabiya-Tobrouq Ridge and the Qunnayyin depression. In fact, the hydrological-hydraulic model which stimulated our investigation, simulates this floodplain with a low to medium probability [14]. However, alkaline seawater is not reconcilable with intermittent supply of freshwater from the floodplain and the absence of cliff incision at the Qunnayyin depression appears to be in contrast to the ~300 mm mean annual coastal precipitation modelled by LOVECLIM for interglacial precession minima periods [15].

Our reconstruction suggests an intermittent stream reaching the Qunnayyin depression from the south and ephemeral discharge from wadi Farigh into the sabkha Kabirah and from there into the sea (Fig 6). Judging from the size of the upper wadi [38] and average flow velocity during a flood event (3 m/s), the bankfull flow could have been ca 600 m³/s which amounts to ca $3 \times 10^9$ m³ bankfull discharge during 60 days of the year. Bankfull discharge from the depression into the wadi could have occurred only during overspill, i.e., when the depression was filled with water. The storage capacity of the empty depression is ca $8 \times 10^6$ m³.

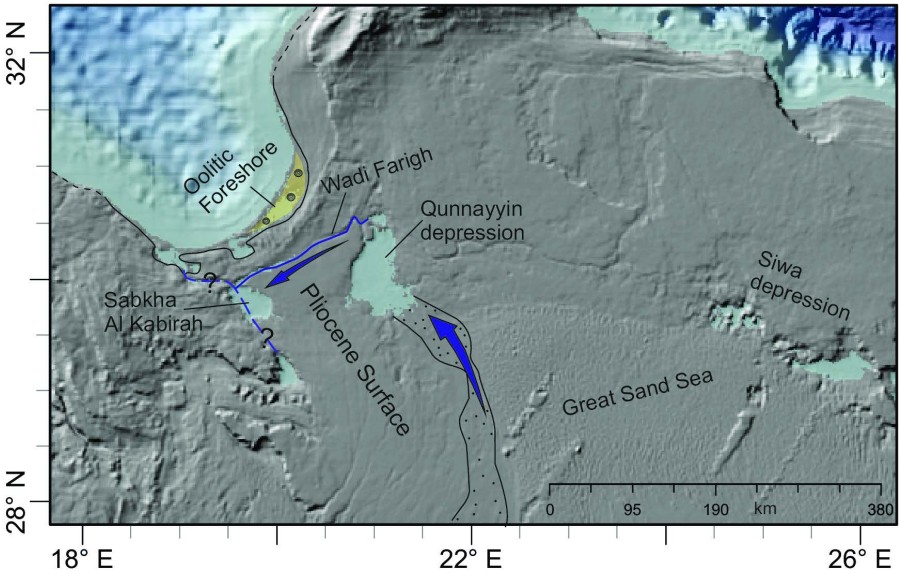

**Fig 6. The water course of the river during MIS 5e reconstructed in this study.** Background data reprinted from NOAA National Centers for Environmental Information 2022: ETOPO 2022 15 Arc-Second Global Relief Model. https://doi.org/10.25921/fd45-gt74 under a CC BY licence.

Two observations support this reconstruction: absence of floodplain deposits in the coastal plain and absence of flood-out deposits in the sabkha Kabirah – both would be due to clear water running during peak discharge events because the Qunnayyin depression served as a sediment trap. However, there is no evidence within the depression for a large-scale sediment trap. Moreover, sabkha Kabirah, situated at ca 25 m a.s.l. and today not connected to the sea, is strikingly free of evaporitic deposits [38] suggesting continuous high groundwater table. Finally, freshwater supplied to the Gulf of Sirt by a large river would have lowered alkalinity of the coastal water and suspended ooid formation. On the other hand, ooids are known for episodic growth and "sleeping" stages [53] so that the cortex growth could have resumed when alkalinity reached required values each time after a peak discharge episode.

There is no definite piece of evidence for or against a river, but the sum of our observations speak for a Saharan transport-limited ephemeral river discharging its water into the Qunnayyin depression where the water recharged the regional aquifer which, in turn, sustained continuous high groundwater table in the coastal plain.

**River-aquifer exchange.** Our reconstruction (Fig 6) suggests that the trans-Saharan river was transport-limited, hence flowing intermittently and this was likely caused by local evapotranspiration and stream bed seepage. Through seepage a suballuvial, unconfined aquifer is typically recharged [54] and, if conduit network permits, its water would reach the local discharge area. On the way to our local discharge area, the Qunnayyin depression, the river must have crossed a south to north rising piezometric surface starting at −50 m a.s.l. at 28°30'N, 21°40'E until reaching 0 m a.s.l at 29°30'N, 21°E [55]. The aquifer's hydraulic conductivity in this area is modelled to ~−5 m/s [56] and the transmissivity is measured to maximum 1500 m²/day in the area around the Qunnayyin depression [56]. This data support the idea of transmission loss, i.e., the river transferred most its water to the local suballuvial aquifer which surfaced in the Qunnayyin area and released its water into the depression. Transmission loss is considered in the hydrological model study [14] as lost water that would not return to the surface. Yet, intermittent to near-perennial river flow resulted from the model simulation implying that rainfall in the source area was perhaps overestimated or the river was partly obstructed and lost its water to local evapotranspiration and stream bed seepage or its course was diverted towards the Kufra basin.

In the Ajdabiya Trough the groundwater table lies above the saline-freshwater interface [57] confirmed by high sodium and chloride concentrations [58,59] with a marked increase in salinity (> 2000mg/l) north of 29 20'N [56]. This should explain the lack of Aterian or other Stone Age sites in the Trough [60–62] as well as the "Great Man-Made River" by which the east Libyan population is today supplied with potable water from the Nubian Sandstone Aquifer in the Kufra basin.

**Other freshwater source: The Chott.** If the Sahabi and Kufrah rivers did not reach the Ionian Sea, the freshwater signal recorded in the deep-sea cores may originate from the Irharhar river via the Chott depression (Tunisia). Elemental ratios and grain-size endmember model data obtained from core CP10 (see Fig 3 for location) point to this area as a key source region [3]. The Chott basins and adjacent coastal zone show features expected to arise from freshwater ponding in the depression and fluvial discharge on the coast. Various deposits in the Chott, albeit understudied, are a testimony for water covering the el Jerid depression in the past (Fig 7). On the coast, the MIS 5 deposits are non-oolitic (Fig 7; [63]) in contrast to the adjacent coastal zone supporting the idea of riverine input into the Gulf during MIS 5. Alluvial fans covering the flanks of the northern Atlas mountains suggest that ephemeral streams existed at the northern border of the Chott basins [64] where they gathered runoff water in addition to lake water. The centres of the Chott basins are situated slightly

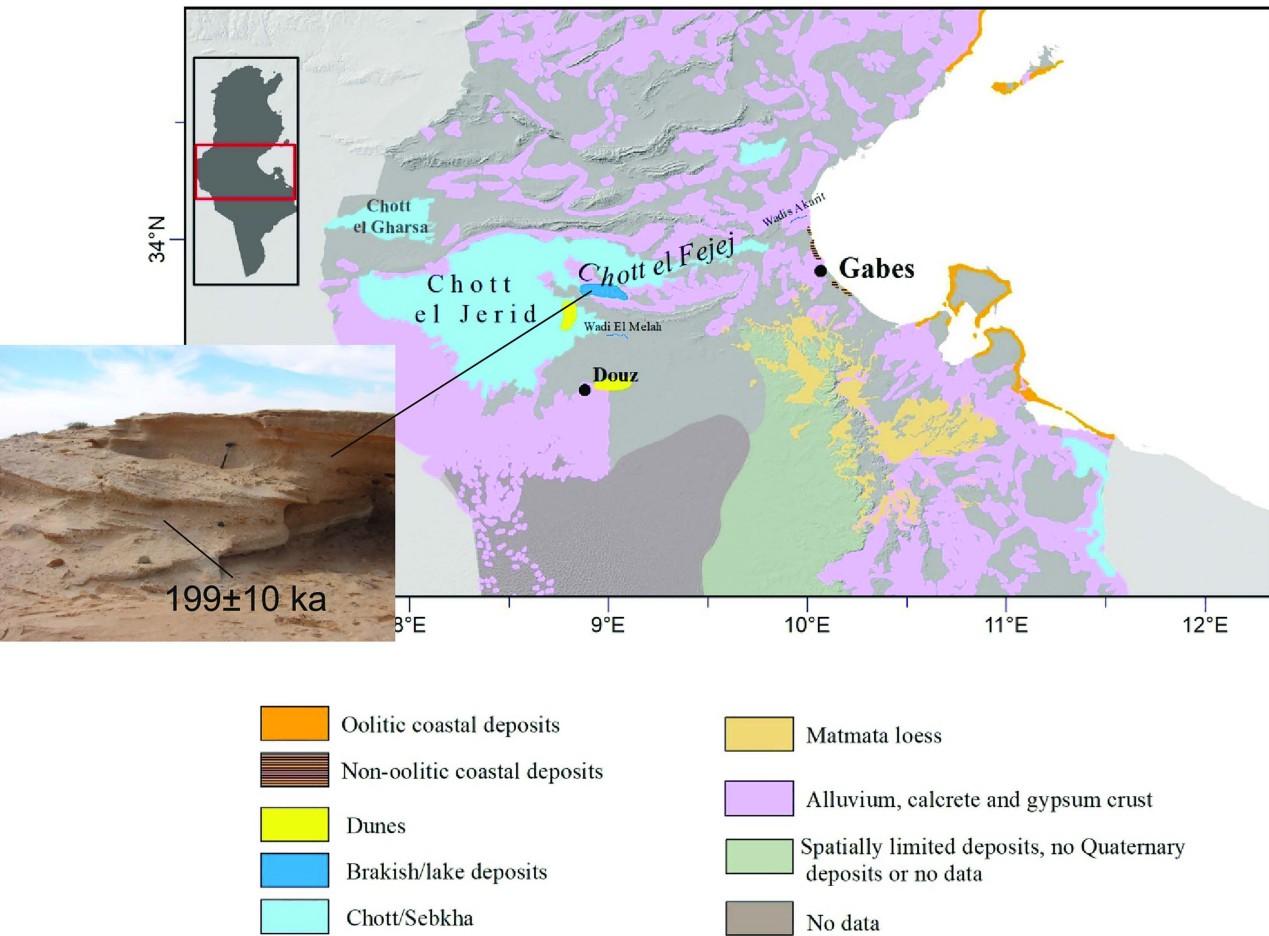

**Fig 7. The Chott basins (Tunisia) and late Quaternary deposits.** Photo inserted shows lake deposit dated to ~200 ka by quartz optical dating (see Supporting Information for details).

above or below modern sea level implying lake outflow occurring when sea level was lower than today and vice versa, saltwater intrusion when sea level was higher than today.

We follow published reasoning [14] that the Chott basins are in a unique position in so far as they gather Atlantic-Mediterranean precipitation and associated runoff from the adjacent Atlas mountains as well as receiving water from the Irharhar river. This points to the mass of water required to generate the geochemical anomalies recorded in the adjacent Ionian Sea.

## Conclusions

Although the deep-sea data clearly suggest a significant flow of freshwater into the Ionian Sea from the Gulf of Sirt, we have not been able to locate the pathway this water took across the Sirt coastal plain. Instead, we find that the river lost most of its water to aquifers and terminated around 200 km inland at the Qunnayyin depression which expanded rapidly to a relatively large size in Quaternary times. The evidence for absence of the allogenic river's downstream part gives rise to speculate about less rainfall in the catchment than quantified from climate models and about diversion of water flow towards another regional base level. Instead of the Gulf of Sirt as dominant source of the freshwater our study points to the Irharhar – Chott pathway.

It is said that the evidence for river courses and lakes is blurred or buried due to migrating sand dunes, flash flood and other short-term high-energy processes which tend to obliterate evidence of fluvial activity. This is certainly true when looking for fluvial activity over the linear length of ca 2000 km, but featured deposits in the Chott basin and northern Sahara show that the desert landscape is rich in evidence for surface water flow, albeit temporary and spatially limited.

Human migration across the Sahara towards the Libyan coast is a plausible conclusion drawn from the idea of Pleistocene green corridors, but the fact that the Sirt coastal plain's aquifer is situated above the marine-freshwater interface suggests that pre-Holocene humans migrated along the Nubian groundwater-discharge path and settled in oases.

## Supporting information

**S1 File. This is the Supporting Information document.**
(PDF)

**S2 File. Inclusivity questionnaire.**
(DOCX)

## Acknowledgments

We are deeply grateful to Prof. Mansour Elbabour who's leadership and commitment made the field work in Libya happening. We thank Mahanad Egwiten who worked out safe and secure field logistics. We thank Veronica Tenczer who helped with generating the figures, in particular the digital geological map.

## Author contributions

**Conceptualization:** Barbara Mauz, Esam Abdulsamad.

**Data curation:** Barbara Mauz, Noureddine Elmejdoub, Moez Mansoura.

**Formal analysis:** Barbara Mauz, Noureddine Elmejdoub.

**Funding acquisition:** Esam Abdulsamad.

**Investigation:** Barbara Mauz, Saleh Emhanna, Moez Mansoura.

**Methodology:** Barbara Mauz.

**Project administration:** Barbara Mauz, Esam Abdulsamad, Saleh Emhanna, Noureddine Elmejdoub.

**Resources:** Esam Abdulsamad.

**Validation:** Barbara Mauz, Saleh Emhanna, Noureddine Elmejdoub, Moez Mansoura, Michael Rogerson.

**Visualization:** Saleh Emhanna, Moez Mansoura.

**Writing – original draft:** Barbara Mauz.

**Writing – review & editing:** Barbara Mauz, Esam Abdulsamad, Noureddine Elmejdoub, Michael Rogerson.

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
