## [Decision Letter · Decision Letter 0]

26 Jun 2025

PONE-D-25-26127Where did the river go? Testing the hypothesis of rivers discharging into the Gulf of Sirt (East Mediterranean) during the late PleistocenePLOS ONE?

Dear Dr. Mauz,

Thank you for submitting your manuscript entitled "Where did the river go? Testing the hypothesis of rivers discharging into the Gulf of Sirt (East Mediterranean) during the late Pleistocene" to PLOS ONE. After careful evaluation by the reviewers and consideration of their comments, I am pleased to inform you that your manuscript is provisionally accepted pending minor revisions. The reviewers have found your study to be of interest and generally sound. However, a few minor points need to be addressed to improve the clarity and rigor of the manuscript. Please carefully consider the reviewers comments and submit a revised version addressing each point in detail.

We look forward to receiving your revised manuscript.

Kind regards,

Rizwan Sarwar Awan

Academic Editor

PLOS ONE

Journal Requirements:

The National Authority for Scientific Research (Libya) funded field work in Libya and laboratory analyses. Field work in Tunisia (Chott area) was supported by Leverhulme Grant IN-2012-113 (awarded to M Rogerson).

6. We note that Figure(s) 1, 2A, 2B, 3, 4A, 5, 6, 7, S3, and S6, in your submission contain [map/satellite] images which may be copyrighted. All PLOS content is published under the Creative Commons Attribution License (CC BY 4.0), which means that the manuscript, images, and Supporting Information files will be freely available online, and any third party is permitted to access, download, copy, distribute, and use these materials in any way, even commercially, with proper attribution. For these reasons, we cannot publish previously copyrighted maps or satellite images created using proprietary data, such as Google software (Google Maps, Street View, and Earth). For more information, see our copyright guidelines: http://journals.plos.org/plosone/s/licenses-and-copyright.

a. You may seek permission from the original copyright holder of Figure(s) 1, 2A, 2B, 3, 4A, 5, 6, 7, S3, and S6 to publish the content specifically under the CC BY 4.0 license.

7. We note that Figure(s) S4, and S6, in your submission contain copyrighted images. All PLOS content is published under the Creative Commons Attribution License (CC BY 4.0), which means that the manuscript, images, and Supporting Information files will be freely available online, and any third party is permitted to access, download, copy, distribute, and use these materials in any way, even commercially, with proper attribution. For more information, see our copyright guidelines: http://journals.plos.org/plosone/s/licenses-and-copyright.

a. You may seek permission from the original copyright holder of Figure(s) S4, and S6 to publish the content specifically under the CC BY 4.0 license.

Reviewers' comments:

Reviewer's Responses to Questions

**Comments to the Author**

1. Is the manuscript technically sound, and do the data support the conclusions?

Reviewer #1: Yes

Reviewer #2: Yes

2. Has the statistical analysis been performed appropriately and rigorously?

Reviewer #1: Yes

Reviewer #2: Yes

3. Have the authors made all data underlying the findings in their manuscript fully available?

Reviewer #1: Yes

Reviewer #2: Yes

4. Is the manuscript presented in an intelligible fashion and written in standard English?

Reviewer #1: Yes

Reviewer #2: Yes

Reviewer #1: The manuscript presents a robust multidisciplinary approach (fieldwork, remote sensing, sedimentological and geochemical analysis) to test the long-debated hypothesis), consider summarizing key data in the main text.

"-" described methods (e.g., sample numbers, thin section analysis) are briefly mentioned and should be described in greater detail for reproducibility.

Optical dating is mentioned, but the key details and uncertainties are only in the supplement. It would help to briefly summarize them in the main text.

The satellite and subsurface data are mentioned, it's not always clear how they integrate with field observations.

line 284 "cortex growth could have been resumed" – this sentence could be clearer.

Some references appear to be repeated or inconsistently cited—please check for duplicates and ensure consistency only where necessary.

The manuscript is an important contribution to Mediterranean paleoenvironmental science and will be suitable for publication after minor revisions

Reviewer #2: The manuscript tests the hypothesis of late Pleistocene rivers discharging into the Gulf of Sirt (Libya) to explain freshwater inputs linked to Mediterranean sapropel formation. Combining field surveys, lab analyses, and literature synthesis, the authors find no evidence for large MIS 5 rivers reaching the Gulf. Instead, they propose ephemeral streams terminating inland (e.g., Qunnayyin depression) and identify the Irharhar-Chott (Tunisia) pathway as a more plausible freshwater source. The study is methodologically robust, addresses a significant paleoclimatic question, and supports alternative mechanisms for sapropel-related hydrology.

Overall: A robust study advancing Mediterranean paleohydrology. Minor edits will amplify its impact.

Revisions Required

Clarify critical evidence:

Explicitly link absence of coastal fluvial deposits (e.g., no incisions/evaporites in Qunnayyin) to the rejection of Sirt rivers. Contrast with Chott’s positive evidence (Fig 7).

Sharpen implications:

Emphasize how the Irharhar-Chott pathway reconciles deep-sea geochemical data (e.g., core CP10) and LOVECLIM rainfall models.

Technical fixes:

Ensure all figures cited (e.g., Fig 6).

Define key terms (e.g., "transport-limited ephemeral river").

Fix minor errors (e.g., duplicate "the" in line 260).

Optional Enhancements

Briefly note how aquifer salinity affects "green corridor" viability near Sirt.

**Do you want your identity to be public for this peer review?** For information about this choice, including consent withdrawal, please see our Privacy Policy

Reviewer #1: **Yes: ** Max Leman

Reviewer #2: **Yes: ** WAEL ABDELWAHAB IBRAHEM

---

## [Author Response · Author response to Decision Letter 1]

13 Aug 2025

please find our responses to reviewers in the file "Response to Reviewers"

There were no specific comments from the editor.

---

## [Editor Report · Decision Letter 1]

20 Aug 2025

Where did the river go? Testing the hypothesis of rivers discharging into the Gulf of Sirt (East Mediterranean) during the late Pleistocene

PONE-D-25-26127R1

Dear Dr. Mauz,

We’re pleased to inform you that your manuscript has been judged scientifically suitable for publication and will be formally accepted for publication once it meets all outstanding technical requirements.

Kind regards,

Rizwan Sarwar Awan

Academic Editor

PLOS ONE
---

## [Editor Report · Acceptance letter]

PONE-D-25-26127R1

PLOS ONE

Dear Dr. Mauz,

I'm pleased to inform you that your manuscript has been deemed suitable for publication in PLOS ONE. Congratulations! Your manuscript is now being handed over to our production team.

Kind regards,

on behalf of

Dr. Rizwan Sarwar Awan

Academic Editor

PLOS ONE